# One-Pot Hydrothermal Preparation of Hydroxyapatite/Zinc Oxide Nanorod Nanocomposites and Their Cytotoxicity Evaluation against MG-63 Osteoblast-like Cells

**DOI:** 10.3390/molecules28010345

**Published:** 2023-01-01

**Authors:** Vignesh Raj Sivaperumal, Rajkumar Mani, Veerababu Polisetti, Kanakaraj Aruchamy, Taehwan Oh

**Affiliations:** 1Department of Biomedical Engineering, PSG College of Technology, Coimbatore 641004, India; 2Department of Physics, PSG College of Arts and Science, Coimbatore 641014, India; 3Wallenberg Wood Science Center, Department of Fibre and Polymer Technology, School of Engineering Sciences in Chemistry, Biotechnology and Health, KTH Royal Institute of Technology, SE-100 44 Stockholm, Sweden; 4School of Chemical Engineering, Yeungnam University, Gyeongsan 38541, Republic of Korea

**Keywords:** hydroxyapatite, zinc oxide, nanocomposite, hydrothermal, biocompatible

## Abstract

In the present study, HAp-ZnO nanorod nanocomposites were successfully prepared using a customized hydrothermal reactor and studied for their compatibility against MG-63 osteoblast-like cells. The crystallinity, morphology, presence of chemical elements, and surface area properties were studied by XRD (X-ray diffraction), FE-SEM (field emission scanning electron microscopy), TEM (transmission electron microscopy), EDS (energy dispersive spectrum) and N_2_ adsorption/desorption isotherm techniques, respectively. Further, the mechanical strength and thermal analysis were carried out using the nanoindentation method and thermogravimetric/differential scanning calorimeter (TG/DSC) methods, respectively. Moreover, in vitro biocompatibility studies for the prepared samples were carried out against human osteosarcoma cell lines (MG-63). The crystalline nature of the samples without any impurity phases was notified from XRD results. The formation of composites with the morphology of nanorods and the presence of desired elements in the intended ratio were verified using FE-SEM and EDS spectra, respectively. The TG/DSC results revealed the improved thermal stability of the HAp matrix, promoted by the reinforcement of the ZnO nanorods. The nanoindentation study ensured a significant enhancement in the mechanical stability of the prepared composite material. Finally, it demonstrated that the HAp matrix’s mechanical strength and thermal stability were improved by the reinforcement of ZnO, and the cytotoxicity evaluation affirmed the biocompatible nature of the biomimetic hydroxyapatite in the composite.

## 1. Introduction

Implementing materials with excellent biological properties integrated with improved mechanical strength has become crucially important for bone implant applications [1]. Numerous calcium phosphates have been researched in the dental and orthopedic areas over the past 20 years, including mono-di calcium phosphate [2], tetra-calcium phosphate [3], amorphous calcium phosphate [4,5], and hydroxyapatite (HAp) [6,7]. The most popular type of bioceramic is HAp because of its close resemblance to the extrinsic mineral that constitutes bone and teeth [8]. Crystallographic and chemical tests have shown that synthetic and naturally occurring HAp are identical and have properties such as being biologically active, nonimmunogenic, osteoconductive, non-inflammatory, non-toxic, and biocompatible [9]. This indicates that it can directly form chemical bonds with biological tissue [10]. Many hydroxyapatite composites have recently been created for a variety of biological applications, including orthopedics and dentistry [11]. 

However, the main drawbacks that prevent HAp from being used more widely in biomedical applications, include their high brittleness, lack of antibacterial properties, and restricted interaction with host tissues [12]. For many HAp-based implants, the absence of bactericidal properties causes prosthetic infection. Numerous studies have found that there has been a large rise in bacterial infections in orthopedic implants over the years. This has resulted in arthroplasty failure and surgical excision of the device, as well as significant physical and psychological trauma for the patients [13]. 

On the other hand, zinc oxide, the widely used antimicrobial agent, has a significant impact on biomedical implants as coating agents, fillers, and as an additive material in scaffolds. The ZnO nanostructures induced osteogenesis and angiogenesis, which played a major role in wound healing and bioactive scaffold development. However, the size and concentration dependent toxicity effect of ZnO material has been a major concern in biomedical applications. It has been reported that the interaction of HAp promotes the chemical transformation of ZnO nanoparticles to zinc phosphate, which reduces the toxicity level of ZnO nanoparticles in the environment [14]. Therefore, the incorporation of ZnO into biomimetic hydroxyapatite to develop a ceramic composite has proven to be an effective material for biomedical applications.

Numerous studies on HAp-ZnO composite-based implant materials demonstrated better compatibility and antimicrobial activity, both in vivo and in vitro [14,15,16,17]. Moreover, in the literature, this composite has been successfully demonstrated as a dental implant coating material with anti-biofilm activity [18], mineralization of osteoblasts [19], antimicrobial activity [20], bone tissue engineering [21], bone implantation [22], improved antibacterial activity and biocompatibility [23], bone recovery [24], sealing ability in endodontically treated teeth [25] and bone defect repair [26].

Therefore, various synthesis methods, such as sol-gel method [27], chemical reduction method [28], chemical solution deposition [29] and co-precipitation method [20], have been used for the preparation of HAp–ZnO composite powders. The hydrothermal method, among others, offers the finished product’s essential physico-chemical properties. The method is carried out in an isolated system, which provides a number of advantages, including quicker interactions between the reactants, a lower operating temperature, enhanced nucleation controls, an increased reaction rate, greater dispersion, greater shape control, and no pollution [30].

However, the preparation of any composite material when using the hydrothermal method typically involves two steps. In the first stage, one of the composite’s component parts is prepared, and the second stage involves combining that component with another part of the composite that is being created in order to prepare the final composite. The drawback is that it requires a lot of time and energy.

The current method can solve this issue because it only requires one step. The second step is eliminated in the hydrothermal reactor’s unique design, allowing for the introduction of another component’s precursor solution while the hydrothermal system is still operating. This sort of investigation is scarce in the literature. Hence, the present study aimed to prepare one-dimensional HAp/ZnO nanocomposites through this uniquely designed hydrothermal reactor, systematically study their physico-chemical, mechanical, and thermal properties, and demonstrate their potentiality and compatibility for bone implant applications.

## 2. Results and Discussion

### 2.1. X-ray Diffraction Analysis

Figure 1 displays the produced HAp-zinc oxide composites and pure HAp’s XRD patterns. Among the XRD data, the HZnO1 (0.125 M of zinc nitrate) composite powder contains major characteristic peaks of HAp (JCPDS File No. 09-0432) and major intensity peaks corresponding to zinc oxide (JCPDS File No. 89-1397). Meanwhile, the composites (HZnO2-HZnO4) exhibit only the major peaks of zinc oxide rather than HAp. This indicates that the increase in molar concentration of zinc nitrate (0.5, 0.75, and 1 M) results in a major distribution of zinc oxide material. In the preparation of composite material, HAp and zinc oxide are considered as matrix and reinforcement materials, respectively. Pure HAp was prepared under optimized hydrothermal conditions, and its XRD data were compared with JCPDS file no. 09-0432, which was used as a reference for the prepared composites (HZnO1-HZnO4). Therefore, the well-characterized HZnO1 composite is further examined for morphological, mechanical, and biological studies.

### 2.2. Electron Microscopy Analysis

The HAp-zinc oxide (HZnO1) composite sample powder was analyzed under an electron microscope for morphological and elemental distribution as shown in Figure 2. The TEM image clearly indicates the formation of composite nanorods with sizes between 30–90 nm in length and breadth in the range of 10–40 nm, as shown in Figure 2a,b. Moreover, the SEM image of the prepared composite shows fine powders without any large aggregate/agglomerate formation (Figure 2c). The elemental mapping clearly illustrates the composite formation of the HAp matrix reinforced with zinc oxide nanoparticles in a uniformly dispersed manner (Figure 2d–h). This was evident from the XRD data, where distinct phases of HAp and ZnO were observed. In addition, the EDS result demonstrates the existence of the desired components, including Ca, P, O, and Zn, respectively, where the obtained Ca/P ratio for the HAp-zinc oxide composite powders was around 1.67 (Figure 2i). The quantitative analysis shows the distribution of ZnO content is around 10% in the composite powder.

### 2.3. TG/DSC Analysis

The TG/DSC graph of the prepared HAp-ZnO composite (HZnO1) is shown in Figure 3. Three stages of weight loss, along with corresponding endothermic peaks (82.8°, 152.8°, and 518.2°), and exothermic peaks (315.4°), were observed. The endothermic peaks at 82.8° and 152.8° correspond to the removal of solvent molecules and volatile surfactant molecules, respectively, which was observed at a 20% weight loss in the TG curve. The exothermic peak at 315.4° corresponds to the pyrolysis of zinc alkoxide, and the endothermic peak at 518.2° corresponds to the crystallization of HAp-zinc oxide composite material. Hence, no considerable weight loss was observed between 600–1000 °C, and the prepared HAp-ZnO composite material was calcined at 650 °C. Moreover, it has been reported that the deformation of the apatite structure with an increase in temperature (>900 °C) causes the lattice replacement of Ca ions with Zn ions [31]. Thus, the calcination process results in the formation of distinct phases of HAp and zinc oxide material, which was evident from the XRD data. The addition of ZnO (10–30 wt.%) into the HAp matrix promotes the thermal stability of HAp without any change in the apatite structure [32]. Thus, the study reveals the characteristics of the HAp with the incorporation of ZnO as an additive material.

### 2.4. BET Analysis

The N_2_ adsorption/desorption graph of the prepared HAp-ZnO composite is shown in Figure 4. It was confirmed that the adsorption and desorption curves belong to a type II isotherm, which is in line with the material’s non-porous characteristics [33,34]. From the graph, the pore diameter, pore volume, and surface area were calculated to be 3.92 nm, 0.025003 cc/g, and 21.827 m^2^/g, respectively. The inset shows the BJH plot of the prepared composite sample, wherein the pore distribution was obtained in a range of 2–25 nm and the peak pore diameter was found at ~5 nm. This demonstrates that the modest surface area obtained could be attributed to the presence of mesopores in the composite material. 

### 2.5. Nanoindentation Study

Using a nanoindentation study, the composite’s mechanical behavior was figured out (Figure 5). The composite powder was made into pellets using a hydraulic pelletizer with a diameter of 12 mm and thickness of 2 mm. The applied pressure was 12 MPa. The load (600 µN) was gradually applied to the pellets using a nanoindenter, and the corresponding displacement was identified. From the load vs. displacement graph for the composite sample, the Young’s modulus and hardness were determined as 4.4 GPa and 0.8 GPa, respectively. The Young’s modulus and hardness value of HAp pellets used as reference material were found to be 3.16 GPa and 0.486 GPa, respectively. This indicates that the incorporation of zinc oxide material has a significant impact on the mechanical behavior of the composite pellets.

### 2.6. In Vitro Biocompatibility

To simulate and forecast biological responses to materials when applied to or on tissues within the body, in vitro cytocompatibility tests were investigated [35]. MG-63 human osteoblast cell cultures, which possess the identical physiology and adhesiveness as human osteoblast cells, were used in the in vitro biocompatibility study to evaluate the prepared composite material [36]. The percentage of cell viability against various concentrations (12.5, 25, 50, 100, 200 µg/mL) of HAp-ZnO composite is shown in Figure 6. The assay clearly shows excellent biocompatibility (~80%) of HAp-ZnO against MG-63 human osteoblast-like cells, regardless of the highest concentration of 200 µg/mL. The MTT experiment therefore amply demonstrates the produced nanocomposite’s high compatibility for orthopedic applications.

## 3. Materials and Methods

Zinc nitrate hexahydrate, calcium nitrate, diammonium hydrogen phosphate, and 25 % ammonia solution were obtained from Merck (Mumbai, India). Without any further treatment, all of the chemicals were utilized. Throughout all reactions, double-distilled water was used. The customized hydrothermal instrument was designed and developed at Amar Equipment Pvt. Ltd. (Mumbai, India). The MTT assay was used in the cytotoxicity investigation using the human osteosarcoma cell lines (MG-63). The National Centre for Cell Science (NCCS) (Pune, India) provided the MG-63 human osteosarcoma cell lines. Fetal bovine serum (FBS) and 10% of Eagle’s minimal essential medium were used to cultivate the cells. The cells were maintained in a CO_2_ incubator at 37 °C, 5% CO_2_, 95% air, and 100% relative humidity. Every week, the medium was switched, while the cultures were switched every other week. The composite powder was applied to the cells in a variety of concentrations (12.5, 25, 50, 100, and 200 µg) for 48 h. Cells that were not exposed to the nanoparticles were kept as control. Using the MTT assay, as mentioned in our previous reports, the percentage of cell viability was evaluated [37,38].

### 3.1. Synthesis of HAp-ZnO Composite

The preparation of HAp-zinc oxide composite powder under a customized hydrothermal instrument proceeds as follows: (i) Initially, the zinc nitrate (0.125 M) was adjusted to pH 10 using a 25% ammonia solution and treated hydrothermally for one hour at 100 °C under a mild stirring rate of 60 rpm. After the process, a freshly prepared mixture containing a Ca/P ratio of 1.67 was added to the reactor. The hydrothermal operation condition was once again maintained at 250 °C for three hours under naturally occurring pressure. Once the reaction was completed, the reactor was cooled down to room temperature, and the obtained powder was calcined at 650 °C for 1 h. A similar process was followed for different concentrations of ZnO, such as 0.125 M (HZnO1), 0.25 M (HZnO2), 0.375 M (HZnO3) and 0.5 M (HZnO4)) maintained at 1.67 for the Ca/P ratio.

### 3.2. Characterization of HAp-ZnO Nanocomposite

Initially, the phase formation was analyzed using X-ray diffraction (XRD, PAN Analytical; Xpert Powder Diffractometer, Eindhoven, Netherlands) to determine the formation of the HAp-zinc oxide composite. The well-formed hydroxyapatite-zinc oxide (HZnO) composite powder was further characterized using thermogravimetry/differential scanning calorimetry (TG/DSC, NETZSCH STA 449F3, Yokohama, Japan), field emission scanning electron microscope (FESEM, SIGMA HV - Carl Zeiss with Bruker Quantax 200 - Z10 EDS Detector, Billerica, USA), transmission electron microscope (TEM, JEOL JEM 2100, Freising, Germany), energy dispersive spectrum (EDS), elemental mapping, Brunauer-Emmett-Teller (BET, BELSOPR-max, Osaka, Japan), nanoindentation study (Hysitron Inc., TI-700 Ubi 1 Scanning Quasistatic Nanoindenter, USA) and invitro biocompatibility study against MG-63 human osteosarcoma cell lines.

## 4. Conclusions

The hydrothermal method proved to be an effective method in the preparation of one-dimensional nanorods with a high aspect ratio, which assists the molecular interaction of the prepared HAp-ZnO composite material in the biological environment. The XRD data illustrate distinct phases of HAp and ZnO materials, which were evident from the elemental mapping of the composite powder. The elemental mapping confirms the sparse distribution of ZnO material in the HAp matrix. The BET analysis revealed the type II isotherm, which helps in relatively high molecular interactions. The addition of ZnO to the HAp matrix has a significant impact on the thermal and mechanical behavior of HAp. Thus, the results demonstrated that ZnO can be considered an additive material for HAp-based biomedical applications.

## Figures and Tables

**Figure 1 molecules-28-00345-f001:**
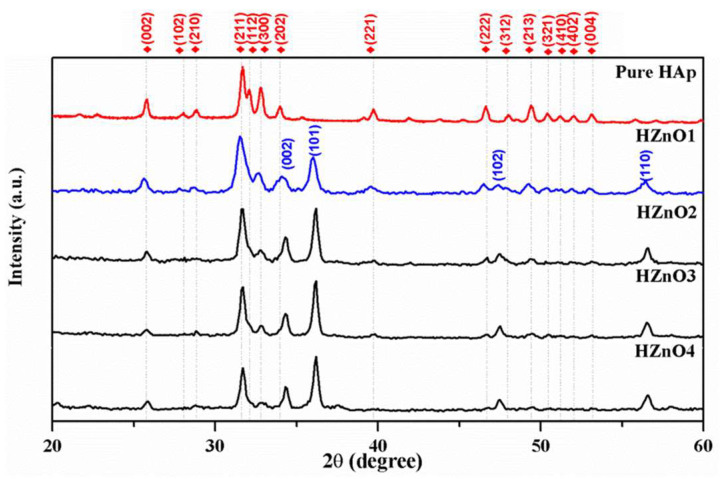
XRD patterns of pure HAp and HAp-ZnO composites.

**Figure 2 molecules-28-00345-f002:**
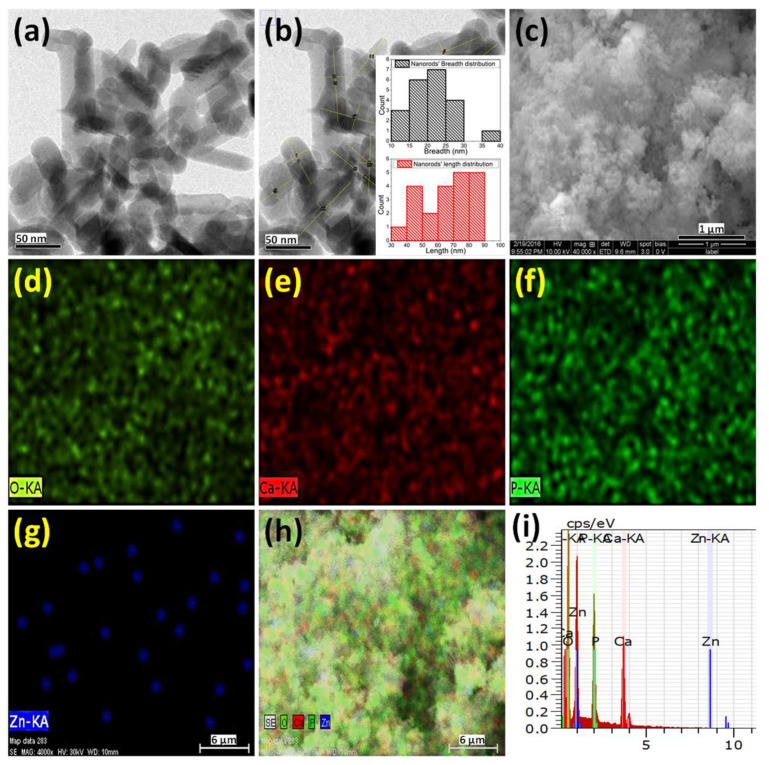
(**a**) TEM (**b**) TEM image with statistical size distribution of the nanorods (**c**) SEM (**d**–**h**) elemental mapping images, and (**i**) EDS spectrum of the prepared HAp-ZnO composite.

**Figure 3 molecules-28-00345-f003:**
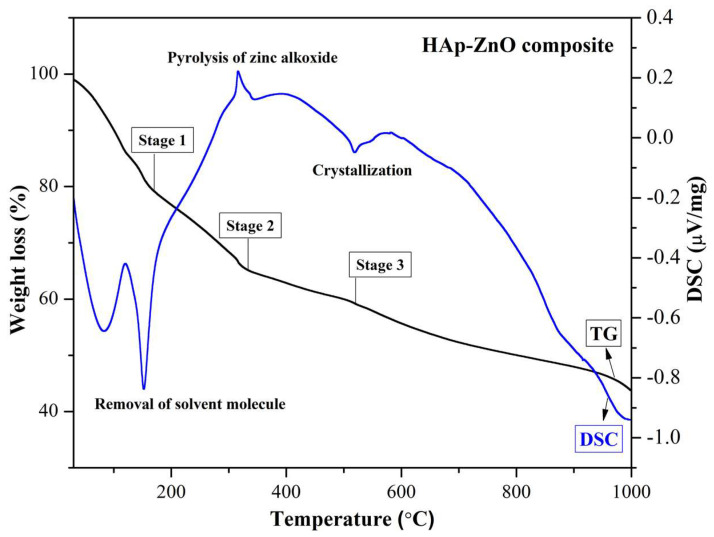
TG/DSC graph of HAp-ZnO composite.

**Figure 4 molecules-28-00345-f004:**
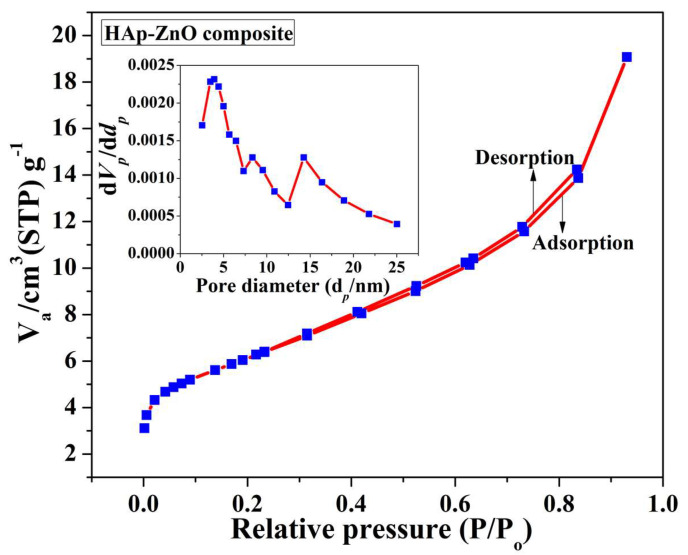
N_2_ adsorption–desorption isotherm of HAp-ZnO composite.

**Figure 5 molecules-28-00345-f005:**
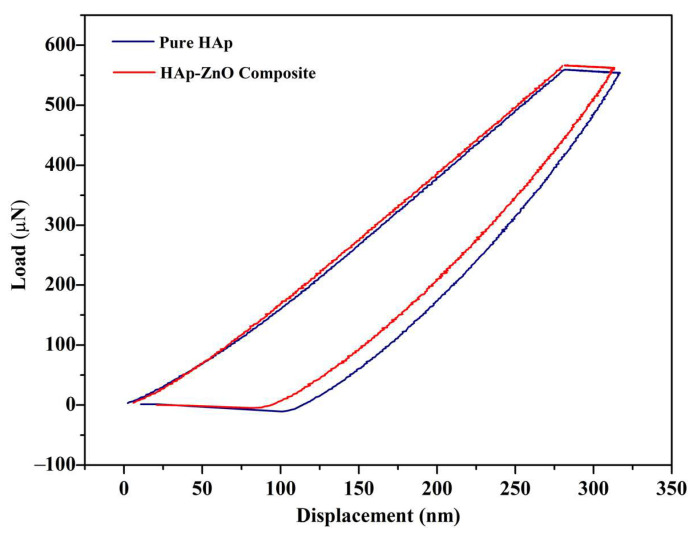
Force–displacement curves of HAp and HAp-ZnO composite.

**Figure 6 molecules-28-00345-f006:**
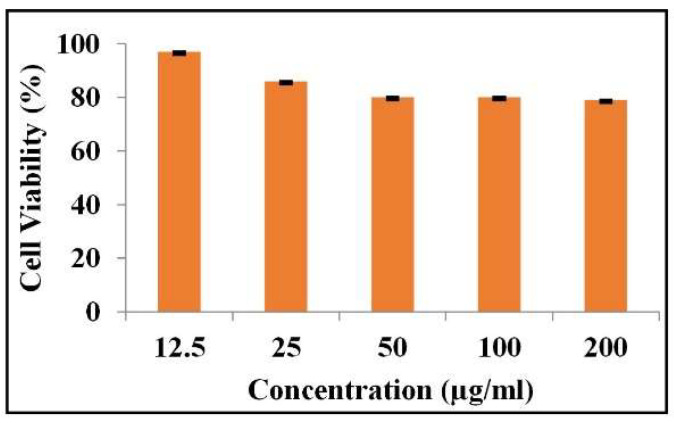
Cell viability of HAp-ZnO composite against MG-63 cell line.

## Data Availability

Not applicable.

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
