# Peer review of "One-Pot Hydrothermal Preparation of Hydroxyapatite/Zinc Oxide Nanorod Nanocomposites and Their Cytotoxicity Evaluation against MG-63 Osteoblast-like Cells"

_molecules, 2023, doi:10.3390/molecules28010345_

Round 1
Reviewer 1 Report
The manuscript is " One-pot hydrothermal preparation of hydroxyapatite/zinc oxide nanorod nanocomposites and their cytotoxicity evaluation against MG-63 osteoblast-like cells ".
General comments:
1. There is no any comment of cytotoxicity by HAp-ZnO in Abstract.
2. No any statements of cell culture in the Methods section.
3. The cytotoxicity evaluation of HAp-ZnO against MG-63 is very poor.
4. This article is the hydrothermal preparation of HAp-ZnO and physical analysis. However, there is only simple analysis, and no in-depth discussion.
This article does not contain any innovations and achievements, which is of little help to readers. Therefore, I think this paper is not suitable for publication.
Author Response
Response to the referees’ comments
Reviewer 1
The manuscript is " One-pot hydrothermal preparation of hydroxyapatite/zinc oxide nanorod nanocomposites and their cytotoxicity evaluation against MG-63 osteoblast-like cells ".
General comments:
- There is no any comment of cytotoxicity by HAp-ZnO in Abstract.
Author response: We thank the reviewer for the valuable comment. The cytotoxicity study has been mentioned in the abstract part of the revised manuscript.
- No any statements of cell culture in the Methods section.
Author response: We thank the reviewer for the valuable comment. As suggested by the reviewer, clear information about the cell culture has been included in the materials and methods section of the revised manuscript.
- The cytotoxicity evaluation of HAp-ZnO against MG-63 is very poor.
Author response: We thank the reviewer for the valuable comment. The cytotoxicity evaluation was carefully carried out for the prepared samples using the MTT assay against MG-63 human osteosarcoma cell lines. Fetal bovine serum (FBS) and 10% Eagle's minimal essential medium was used to cultivate the cells. The cells were maintained in a CO2 incubator at 37 °C, 5% CO2, 95% air, and 100% relative humidity. Every week, the medium was switched, while the cultures were switched every other week. The composite powder was applied to the cells in a variety of concentrations (12.5, 25, 50, 100, and 200 µg) for 48 hours. Cells that were not exposed to the nanoparticles were kept as control. Using the MTT assay, as mentioned in our previous reports, the percentage of cell viability was evaluated (Materials characterization 134 (2017): 416-421; Applied Sciences 12.21 (2022): 11056).
- This article is the hydrothermal preparation of HAp-ZnO and physical analysis. However, there is only simple analysis, and no in-depth discussion.
This article does not contain any innovations and achievements, which is of little help to readers. Therefore, I think this paper is not suitable for publication.
Author response: We respect the valuable comments from the reviewer. The preparation of any composite material when using the hydrothermal method typically involves two steps. In the first stage, one of the composite's component parts is prepared, and the second stage involves combining that component with another part of the composite that is being created in order to prepare the final composite. The drawback is that it requires a lot of time and energy. The current method can solve this issue because it only requires one step. The second step is eliminated in the hydrothermal reactor's unique design, allowing for the introduction of another component's precursor solution while the hydrothermal system is still operating. Furthermore, this is an alternative time-saving approach for synthesizing HAp/ZnO composite, which is a critical step for this product to be used in industrial applications in the future. This sort of investigation is scarce in the literature. Hence, the present study aimed to prepare one-dimensional HAp/ZnO nanocomposites through this uniquely designed hydrothermal reactor, and systematically study their physico-chemical, mechanical, and thermal properties and demonstrate their potentiality and compatibility for bone implant applications.

Reviewer 2 Report
In this manuscript, the author developed Hap-ZnO nanorod via customized hydrothermal reactor. The Hap-ZnO nanorods were characterized by XRD, FE-SEM, and EDS. The bioavailability was evaluated by MG-63 osteoblast-like cells. In the introduction section, the authors provide comprehensive introduction of HAp-ZnO materials. However, there are some concerns that need to be addressed.
1. There is lack of description of the cell culture method and MTT method, please add the methods description.
2. In Figure 2, the labels of scale bar are hard to read. Please make the label clear.
3. In figure 2i, the information of the figure was shielded by label i. Please correct this.
4. The size of nanorods were only detected by TEM images but without statistic evaluation from the images, please add the statistic particle size of TEM images. In addition, please use other method such as DLS measure the size of the nanorods.
5. There is lack of stability evaluation of the HAp-ZnO, please add ZnO release studies to detect the stability of HAp-ZnO nanorods.
Author Response
Response to the referees’ comments
Reviewer 2
In this manuscript, the author developed Hap-ZnO nanorod via customized hydrothermal reactor. The Hap-ZnO nanorods were characterized by XRD, FE-SEM, and EDS. The bioavailability was evaluated by MG-63 osteoblast-like cells. In the introduction section, the authors provide comprehensive introduction of HAp-ZnO materials. However, there are some concerns that need to be addressed.
- There is lack of description of the cell culture method and MTT method, please add the methods description.
Author response: We thank the reviewer for the valuable comment. As suggested by the reviewer, clear information about the cell culture and MTT method has been included in the materials and methods section of the revised manuscript.
- In Figure 2, the labels of scale bar are hard to read. Please make the label clear.
Author response: Figure 2 has been modified and included in the revised manuscript for the improvement of clarity.
- In figure 2i, the information of the figure was shielded by label i. Please correct this.
Author response: We thank the reviewer for the valuable comment. As suggested by the reviewer the corrections have been included in Figure 2 of the revised manuscript.
- The size of nanorods were only detected by TEM images but without statistic evaluation from the images, please add the statistic particle size of TEM images. In addition, please use other method such as DLS measure the size of the nanorods.
Author response: We thank the reviewer for the valuable comment. As suggested by the reviewer the statistic particle size of the as-prepared nanorods was calculated using ImageJ software and included in the revised manuscript (Figure 2b). And, we regret for not being able to provide the DLS measurement data due to the in-accessibility of the facilities presently.
- There is lack of stability evaluation of the HAp-ZnO, please add ZnO release studies to detect the stability of HAp-ZnO nanorods.
Author response: We thank the reviewer for the valuable comment. The present work mainly focused on the preliminary studies for the preparation of HAp-ZnO composites using a specially designed hydrothermal reactor, and its further cytotoxicity evaluation. Hence, the ZnO release study was not carried out. However, we will consider the reviewer’s valuable comment and incorporate it into the extension of this work in the future.

Round 2
Reviewer 1 Report
Except of the cytotoxicity of MG63 by HAp-ZnO, I recommend to do more experiments of HAp-ZnO on the other behavior of MG63.
Author Response
Response to the referees’ comments
Reviewer 1
Except of the cytotoxicity of MG63 by HAp-ZnO, I recommend to do more experiments of HAp-ZnO on the other behavior of MG63.
Author response: We thank the reviewer for the valuable comment. The present work focuses mainly on the preliminary studies for the preparation of HAp-ZnO composites using a specially designed hydrothermal reactor, and its further cytotoxicity evaluation. Hence, only the cytotoxicity studies against MG-63 human osteosarcoma cell lines for the prepared HAp-ZnO composite samples were carried out in the present work. However, we will consider the reviewer’s valuable comment to do more experiments on the other behavior of MG63 and incorporate it into the extension of this work in the future.
Reviewer 2 Report
The revised version addressed all of my concerns.
Author Response
We thank the reviewer for the kind recommendation of the manuscript for publication. As suggested by the reviewer, the manuscript was carefully checked for grammatical corrections and English improvement, and the changes have been included in the revised manuscript.